# A stakeholder engagement strategy for an ongoing research program in rural dementia care: Stakeholder and researcher perspectives

Debra Morgan [1¤]*, Julie Kosteniuk[1], Megan E. O'Connell[2], Norma J. Stewart[3], Andrew Kirk[4], Allison Cammer[5], Vanina Dal Bello-Haas[6], Duane P. Minish[1], Valerie Elliot[1], Melanie Bayly[1], Amanda Froehlich Chow[7], Joanne Bracken[8], Edna Parrott[9], Tanis Bronner[10]

**1** Canadian Centre for Health & Safety in Agriculture, Department of Medicine, University of Saskatchewan, Saskatoon, Saskatchewan, Canada, **2** Department of Psychology, University of Saskatchewan, Saskatoon, Saskatchewan, Canada, **3** College of Nursing, University of Saskatchewan, Saskatoon, Saskatchewan, Canada, **4** Department of Medicine, Neurology Division, University of Saskatchewan, Saskatoon, Saskatchewan, Canada, **5** College of Pharmacy and Nutrition, University of Saskatchewan, Saskatoon, Saskatchewan, Canada, **6** School of Rehabilitation Science, McMaster University, Hamilton, Ontario, Canada, **7** School of Public Health, University of Saskatchewan, Saskatoon, Saskatchewan, Canada, **8** Alzheimer Society of Saskatchewan, Regina, Saskatchewan, Canada, **9** Family Caregiver, Yorkton, Saskatchewan, Canada, **10** Saskatchewan Health Authority, Tisdale, Saskatchewan, Canada

¤ Current address: Canadian Centre for Health & Safety in Agriculture, University of Saskatchewan, Saskatoon, SK, Canada

* debra.morgan@usask.ca

**Data Availability Statement:** All relevant data are within the paper and its Supporting Information files (S5 and S6).

## Abstract

Participatory research approaches have developed in response to the growing emphasis on translation of research evidence into practice. However, there are few published examples of stakeholder engagement strategies, and little guidance specific to larger ongoing research programs or those with a rural focus. This paper describes the evolution, structure, and processes of an annual Rural Dementia Summit launched in 2008 as an engagement strategy for the Rural Dementia Action Research (RaDAR) program and ongoing for more than 10 years; and reports findings from a parallel mixed-methods study that includes stakeholder and researcher perspectives on the Summit's value and impact. Twelve years of stakeholder evaluations were analyzed. Rating scale data were summarized with descriptive statistics; open-ended questions were analyzed using an inductive thematic analysis. A thematic analysis was also used to analyze interviews with RaDAR researchers. Rating scale data showed high stakeholder satisfaction with all aspects of the Summit. Five themes were identified in the qualitative data: hearing diverse perspectives, building connections, collaborating for change, developing research and practice capacity, and leaving recharged. Five themes were identified in the researcher data: impact on development as a researcher, understanding stakeholder needs, informing research design, deepening commitment to rural dementia research, and building a culture of engagement. These findings reflect the key principles and impacts of stakeholder engagement reported in the literature. Additional findings include the value stakeholders place on connecting with stakeholders from diverse

**Funding:** Funding for this research was provided to author DM by the Bilokreli Family Trust Fund (fund #417218), and the Saskatchewan Health Research Foundation through a partnership with the Canadian Institutes of Health Research (authors DM, JK, MEO, NJS, AK, AC, VDBH, JB) in support of the Canadian Consortium in Neurodegeneration in Aging (CCNA, https://ccna-ccnv.ca/) (grant number 049-53). The funders had no role in study design, data collection and analysis, decision to publish, or preparation of the manuscript.

**Competing interests:** The authors have declared that no competing interests exist.

backgrounds, how the Summit was revitalizing, and how it developed stakeholder capacity to support change in their communities. Findings indicate that the Summit has developed into a community of practice where people with a common interest come together to learn and collaborate to improve rural dementia care. The Summit's success and sustainability are linked to RaDAR's responsiveness to stakeholder needs, the trust that has been established, and the value that stakeholders and researchers find in their participation.

## Introduction

With the shift toward implementation and translation of research evidence into practice, particularly in the health sciences, there has been a corresponding emphasis on more participatory research approaches or co-construction of research by researchers and those interested in or affected by the issue being studied. Engaged approaches have emerged in response to increasing awareness of the need for research to be linked to a change process that will make a difference to communities [1]. Stakeholder engagement involves an iterative process of actively seeking the knowledge and experience of a broad range of individuals with a direct interest in an issue, to create a shared understanding and make decisions [2]. Evidence supporting the value of stakeholder engagement for stakeholders, researchers, and the research process is growing and includes improved relevance of the research, quality of the research process, interpretation of findings, and stakeholder outcomes such as increased confidence and sense of personal achievement [3–10]. Active and meaningful engagement of patients and other stakeholders as partners in health research is increasingly required by funding agencies in many countries as a key strategy to close the research-practice gap.

Community-based participatory research (CBPR) is a collaborative research approach involving stakeholders and researchers as equal partners in addressing issues of importance to the community [11]. CBPR aims to combine knowledge with social action and change, with a focus on eliminating health disparities [12] using a collaborative process that values stakeholders' strengths and contributions [13]. Key features of CBPR include building stakeholder capacity for meaningful and equitable participation, and active engagement of stakeholders throughout the research process with the aim of utilizing their knowledge [1, 12]. Stakeholders include individuals, groups, organizations, or communities that have a direct interest in the processes and outcomes of the research partnership [2, 14]. There is a movement toward university-community partnerships as their value in addressing current health challenges has gained recognition [15].

Although the body of literature on models and guidelines for stakeholder engagement in CBPR is growing [8, 14, 16–18], there are few published examples of engagement strategies, descriptions of how they are implemented, or their outcomes [6, 8, 19]. Moreover, most of the recommendations and guidelines on stakeholder engagement are designed for a specific research project and there is little guidance available for researchers seeking to involve stakeholders in a larger ongoing research program. Slunge et al. [20] suggest that developing a stakeholder interaction strategy for a broader research program can help to inform strategic planning, influence policy development, share research knowledge, and improve the quality of research. However, little is known about stakeholder or researcher perceptions and experiences of being involved in such longer-term engagement strategies. As well, most studies have focused on impacts of stakeholder engagement on the research, while much less is known about stakeholder perspectives and stakeholder outcomes. A synthesis of the engagement

literature found that few studies adequately described the engagement context, processes, and impacts [5]. There are also few studies exploring how to implement collaborative research approaches in rural or remote communities that are distant from research centres [21, 22].

The current paper addresses identified gaps in the stakeholder engagement literature by: 1) describing the evolution, structure, and processes of a stakeholder engagement strategy for an ongoing rural dementia care research program; and 2) reporting an evaluation of the engagement strategy, including the perspectives of both stakeholders and researchers on its value and impact.

## The Rural Dementia Action Research program

The Rural Dementia Action Research (RaDAR) program was launched in 1997 by a university-based research team located in a mid-western province in Canada, with the aim of improving health service delivery for people with dementia and their caregivers living in rural and remote settings [23]. The province has a population of 1,098,352 in an area 651,035 km$^2$ and a population density of 1.9 persons/km$^2$. A larger proportion of the provincial population (39%) is rural (living in areas with less than 10,000 people) compared to 19% rural in the rest of Canada [24]. Almost half of the population lives in two major urban centres. The low population density and concentration of specialists in urban centres create challenges in access to dementia-specific services and programs in rural settings [25, 26]; and a shortage of physicians in rural areas of the province impacts access to primary care services [27].

Community-based participatory research methods have guided the program from the beginning and have become more embedded as the benefits became evident and approaches for researcher-stakeholder engagement have advanced (see Morgan et al., 2014 [28] for the evolution of CBPR in the RaDAR program). The interdisciplinary RaDAR team includes researchers and clinicians from nursing, clinical psychology, medicine, psychiatric epidemiology, physical therapy, and nutrition. Individual team members lead their own projects in collaboration with RaDAR and other investigators. Examples of CBPR projects conducted under the RaDAR umbrella include the design, implementation, and evaluation of rural dementia interventions such as a specialist rural and remote memory clinic [29], telehealth support groups for spouses of individuals diagnosed with atypical dementias in a specialist memory clinic [30], and rural primary care-based memory clinics [31, 32].

## The Rural Dementia Summit

The Summit was initiated in 2008 as a stakeholder engagement strategy to fulfill the requirement of continuous knowledge user involvement for a 5-year research grant to support the RaDAR program. Our approach was to create a 27-member decision-maker advisory council drawing on multi-sector relationships developed over the previous decade. Council members deemed that in addition to more traditional, passive engagement strategies (RaDAR website, newsletters, annual reports), an annual face-to-face meeting was critical for active engagement. Thus, the annual Rural Dementia Summit was launched in an urban centre that is centrally located for those attending from across the province. Both stakeholders and researchers recognized the importance of the Summit to dementia care research in the province, thus the Summit has continued to be held annually. The original advisory council has expanded to include a diverse range of stakeholders from across the province that is known as the Knowledge Network in Rural and Remote Dementia Care.

Although the Summit was initiated as an engagement strategy for the RaDAR program, we learned that bringing stakeholders together regularly created the opportunity to share information on other rural dementia research and best practices, which is not easily accessible to rural

stakeholders. Our goals have evolved from the original objective of creating an overarching engagement strategy for a CBPR research program, to also include opportunities for stakeholders to learn about and engage with other current dementia research and programs. This expansion supports the capacity-building element of CBPR by providing stakeholder exposure to, and engagement in, a broad spectrum of rural dementia care and research.

**Participants and Summit organization.** Details about Summit stakeholder participants, organization, funding, venue, evaluation forms, and evolution in response to stakeholder feedback are reported in **S1 Table**. The Summit is organized as an invited event to ensure broad representation of stakeholders from different sectors and geographic areas of the province. The event was held in-person from 2008–2019, and virtually in 2020–2022 due to the Covid-19 pandemic. Invitations are sent to members of the Knowledge Network who have attended past Summits (range 220–250 individuals in years 2016–2019), which includes individuals living with dementia and family members, a diverse range of health care providers and administrators, Ministry of Health representatives, Alzheimer Society staff and leadership, and others. We accommodate requests from other individuals and organizations interested in attending Summit. **Fig 1** illustrates the number of new vs. returning Summit participants over 12 years, while **Fig 2** shows the composition of participants over time.

The Summit begins with an evening poster session followed by a full day of interactive sessions and presentations (**S1 Table**). The poster session is intended to welcome stakeholders to Summit and provide an opportunity for networking and engaging with poster presenters in a relaxed atmosphere. Appetizers are provided, and the room is set up to accommodate mingling and conversations. Typically, 20–25 posters are presented by RaDAR researchers, trainees, other researchers, and community-based organizations providing dementia services. On the following day an introductory presentation is followed by the core activity of an interactive

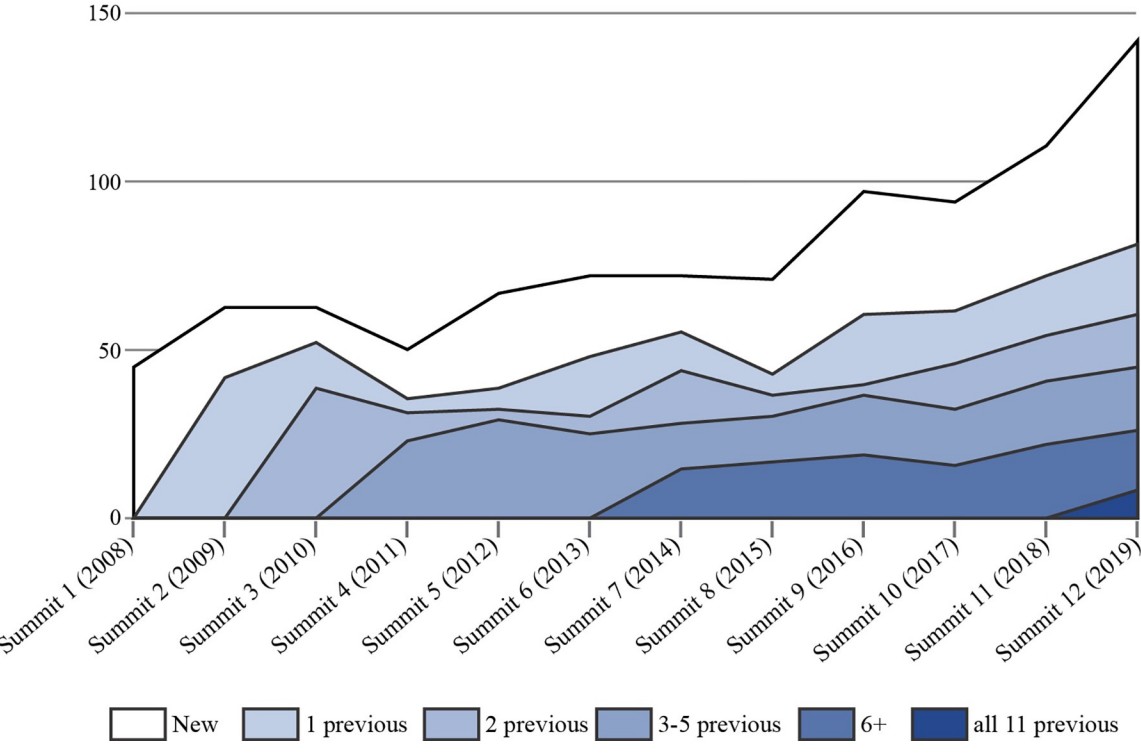

**Fig 1. New and returning Summit participants by year.**

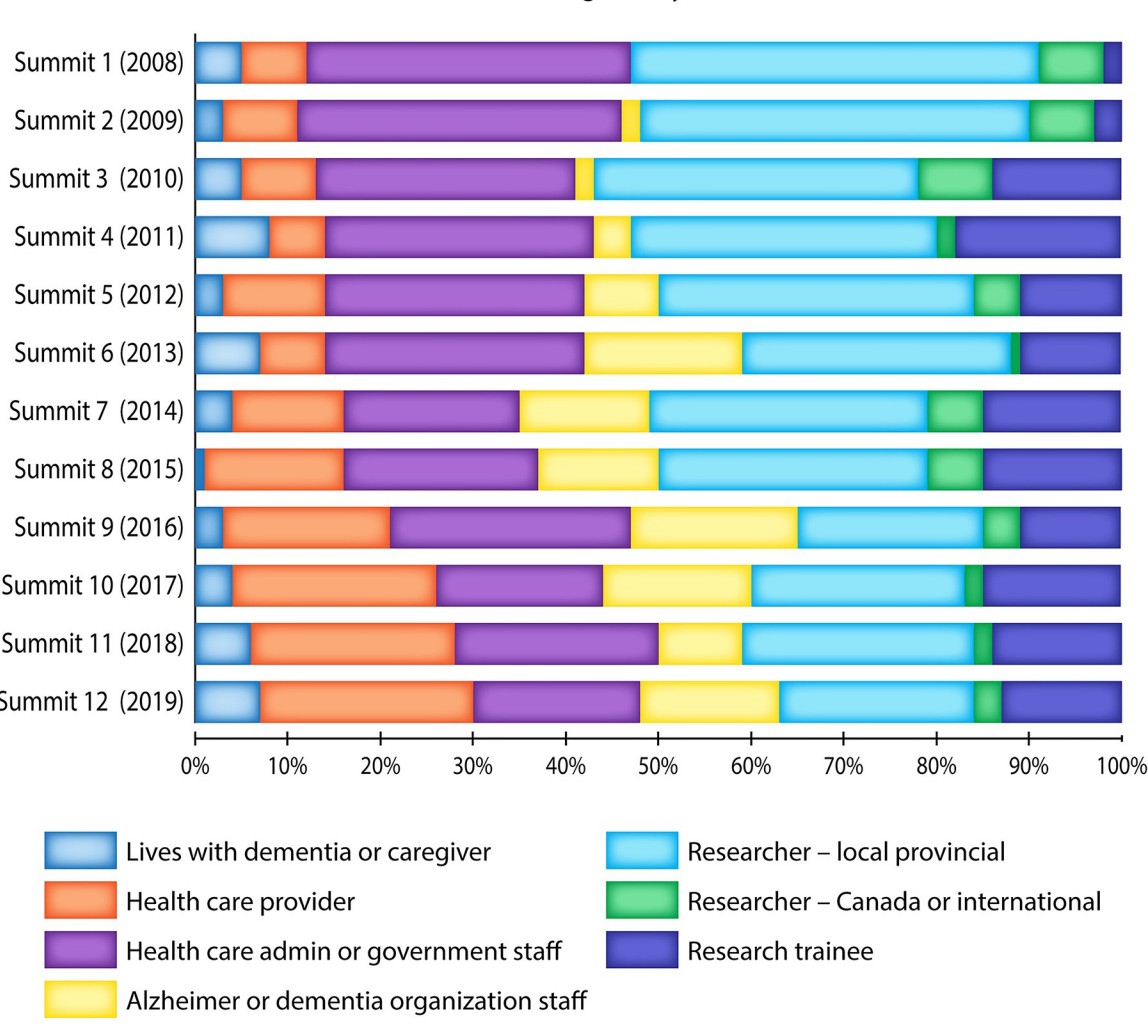

**Fig 2. Summit participants by category by year.**

small group session focused on engaging stakeholders in a new or in-progress RaDAR study, to identify and refine research questions, design study methods, or plan dissemination strategies (see **S1 File** for the focus and purpose of the small group sessions each year). Then a facilitated discussion is held that involves all groups (see **S2 Table** for an example from Summits 5–7). Other components of Summit (e.g., RaDAR research highlights, panels on community-led projects, hearing from those with lived experience with dementia, Alzheimer Society program update) also include opportunities for engagement. These sessions complement other engagement activities for individual RaDAR projects conducted outside of the Summit, by supporting broad stakeholder input that would not be feasible within the resources of individual projects. The differences between Summit and a typical scientific conference are reported in **S3 Table**.

## Methods

The study aims were to describe the evolution, structure, and processes of the annual Summit as a long-term engagement strategy, measure stakeholder satisfaction with various

components of the Summit, and to evaluate stakeholder and researcher perspectives of the Summit's value and impact. A parallel mixed-methods (QUAL + QUAN) approach [33] was used to gather quantitative and qualitative data to address these research questions, using a paper-based evaluation completed by Summit stakeholder participants each year over 12 years (2008–2019) and semi-structured focus groups (2017). Qualitative data were also collected using a focus group and interviews with RaDAR team members at 12 years (2020). This approach allowed an exploration of diverse perspectives and provided a comprehensive picture of the Summit as an ongoing engagement strategy.

## Stakeholder evaluation data collection

Since Summit 1 in 2008, participants have completed an anonymous paper-based evaluation at the end of the day that includes ratings of various aspects of the poster session and Summit day and open-ended questions (see **S2 File** for stakeholder data). Demographic information such as age, sex, and gender were not collected because the number of participants is small and demographic information, in addition to their role, could be identifying. Over the years, additional evaluation questions have been added, some carried forward, and some discontinued. A list of questions by year (Summit 1–12) is shown in **Table 1**. Although the evaluations were deemed exempt from ethics approval by the University Behavioral Research Ethics Board because it was a program evaluation, participants were informed that the data could be used for research publications.

For the poster session participants are asked about their opportunity to interact with researchers, learn about rural dementia research, and receive good value for their time (yes/no for Summits 1–8 and 5-point scale of excellent, very good, good, fair, and poor for Summits 9–12). An open-ended comments section is provided. For the Summit Day, participants rate their agreement on a 4-point scale (strongly agree, agree, disagree, strongly disagree) with statements about time allotted for agenda items and networking breaks, flow of events, ability to share their ideas, and value for time. Responses were rated on a 4-point scale for Summit 1 (very satisfied, somewhat satisfied, could be better, definitely not satisfied), a 4-point scale for Summits 2–8 (strongly agree, agree, disagree, strongly disagree), and a 5-point scale for Summits 9–12 (extremely satisfied, very satisfied, somewhat satisfied, satisfied, not satisfied). Open-ended questions examined what participants liked most, what could be improved, and suggestions for the next Summit. Starting with Summit 9, questions about impact were added, such as what Summit provides that they have not received elsewhere, whether costs to their organization have been returned, and whether practice changes have resulted from attending. At Summit 10, the small group sessions were used to reflect on the value of Summit as a stakeholder engagement strategy and to develop strategies for maximizing engagement going forward (**S2 Table**). Transcripts of these sessions were analyzed for this paper.

## RaDAR team member data collection

RaDAR team perspectives were collected by focus group (6 members) and email interviews (8 members) in 2020 (see **S3 File**). These included 7 faculty, all of whom attended all years of Summit; 6 trainees who attended 1 to 5 years; and 1 research staff who attended all years. Team members were asked how being involved in the Summit as a presenter and as a participant has influenced them, their research, and the RaDAR program more broadly.

## Analysis

Descriptive statistics (frequencies) were used to summarize the results of stakeholder rating scale data. Poster event data are expressed as the percentage of positive responses, specifically

**Table 1. Poster session and Summit day evaluation items: Summit 1 (2008) to Summit 12 (2019).**

| | Summit where item was used[a] | % of positive responses across Summit years[b,c] |
|---|---|---|
| **POSTER SESSION EVALUATION ITEMS** | | |
| *Likert Scale questions*: | | |
| • Opportunity to learn about rural dementia care research | 1–12 | 96–100 |
| • Opportunity to interact with others interested in dementia care | 4–12 | 97–100 |
| • Opportunity to interact with research team members | 1–8 | 97–100 |
| • Provide good value for your time | 1–12 | 95–100 |
| • Overall quality of the posters | 9–12 | 97–100 |
| • Would you recommend the poster session to a friend? | 10 | 100 |
| • Venue | 9–12 | 95–100 |
| • Co-hosting with the provincial Alzheimer Society | 12 | 100 |
| *Open-ended questions (data extracted for thematic analysis)* | | |
| • Comments on the poster session | 1–11 | n/a |
| **SUMMIT DAY EVALUATION ITEMS** | | |
| *Likert scale questions*: | | |
| • Time allotment for agenda items | 1–8 | 84–100 |
| • Time allotment for breaks | 1–8 | 92–100 |
| • Order and flow of day's activities | 1–10 | 98–100 |
| • Opportunity to share your opinions and ideas | 1–12 | 95–100 |
| • It was worth my time to participate in this Summit | 1–12 | 95–100 |
| • Opportunity to increase knowledge of rural/remote dementia care | 9–12 | 100 |
| • Satisfaction with the small group session | 9–12 | 86–100 |
| • Satisfaction with materials provided for the meeting | 9–12 | 96–100 |
| • Satisfaction with keynote presentation | 11–12 | 100 |
| • Satisfaction with Alzheimer Society update | 11–12 | 100 |
| • Satisfaction with Lived Experience Panel | 11–12 | 100 |
| • Satisfaction with Panel on community-led projects | 11–12 | 98–99 |
| • Satisfaction with Summit venue | 1–12 | 94–100 |
| *Open-ended questions (data extracted for thematic analysis)* | | n/a |
| • The elements that you liked best were | 2–11 | |
| • The elements that you liked least or could be improved | 2–11 | |
| • For next year's Summit, you would make sure that | 2–11 | |
| • Next year a topic or presenter that you would include would be | 5–11 | |
| • What Summit provided that you haven't gotten elsewhere | 9–11 | |
| • The costs to your organization have been returned? | 9–11 | |

*(Continued)*

**Table 1.** (Continued)

|  | Summit where item was used[a] | % of positive responses across Summit years[b,c] |
|---|---|---|
| • What changes in your work do you think you will make? | 9–11 | |
| • What made you want to attend Summit? | 10 | |
| • One concrete thing that you will take away? | 10 | |
| • Is there anything in particular missing at this Summit? | 10 | |
| • Thinking ahead, what would be your vision for this event? | 10 | |

[a]Summit 1 (2008) to Summit 12 (2019).

[b]Poster event data represent % of positive responses: "yes" (Summits 1–8) and "excellent/very good/good" (Summits 9–12).

[c]All Summit day Likert scale data represent % of positive ratings (e.g., "extremely/very satisfied")

'yes' (Summits 1–8) and 'excellent/very good/good' (Summits 9–12). Summit Day scales were dichotomized and the percentage of positive ratings are presented (i.e., very/somewhat satisfied, strongly agree/agree, extremely/very satisfied). For open-ended questions, responses to each question were combined across the 12 years and an inductive thematic analysis [34] was conducted to identify themes related to the Summit as an engagement strategy for an ongoing research program. NVivo software [35] was used to organize the qualitative data and facilitate thematic coding. All members of the analytic team (DM, VE, JK, MB, DPM) have been involved in Summit for many years. Analysis steps involved repeated reading of the data to generate initial codes, organizing codes into themes, confirming that themes fit with the full data set, refining the themes, and final analysis [34]. Initial coding and development of themes was conducted by VE and DM, then independently reviewed by JK, MB, and DPM to refine the analysis. Thematic analysis of RaDAR team member data was conducted by DM using the same steps, then refined by members of the analytic team.

# Results

## Stakeholder perspectives

The number of participants attending the Summit each year ranged from 47 to 136 and with the exception of Summit 2 (35%) the percentage of evaluations returned each year ranged from 47% to 83%. Averaged across all years, participants described their role as working directly with people with dementia (42%), working in the field of dementia care at the administrative level (15%), researcher (16%), person living with dementia or a family member (12%), student (9%), or other (17%) (Table 2).

Table 1 shows the range of positive responses such as "extremely/very satisfied" and "strongly agree/agree" for rating scale questions across the 12 Summits. For example, for "opportunity to interact with others interested in dementia care" the percentage of positive responses ranged from 97% to 100% across Summit years. With respect to the poster evening, the majority of participants each year were satisfied with the opportunity to interact with others interested in dementia care, learn about rural dementia research, the value they received for their time, the quality of the posters, and the venue. Responses for the Summit Day were also positive, with the majority very satisfied with opportunities to share their ideas and increase their knowledge of rural dementia care, and the timing and flow of the day.

**Table 2. Summit participant numbers and role by Summit year.**

| Summit Year[a] | Total evaluations completed (n) | Total attending (n) | Response rate (%) | Participant role[b] (%) | | | | | |
|---|---|---|---|---|---|---|---|---|---|
| | | | | Researcher | Student | Work directly with people with dementia in rural area | Person living with dementia or family member | Work at administrative level in dementia care | Other[c] |
| 1 * | 32 | 43 | 74 | * | * | * | * | * | * |
| 2 * | 21 | 60 | 35 | * | * | * | * | * | * |
| 3 * | 28 | 60 | 47 | * | * | * | * | * | * |
| 4 | 40 | 48 | 83 | 30 | 13 | 30 | 10 | 7 | 13 |
| 5 | 31 | 64 | 48 | 19 | 13 | 42 | 10 | 13 | 16 |
| 6 | 40 | 69 | 58 | 8 | 3 | 43 | 5 | 23 | 30 |
| 7 | 38 | 69 | 55 | 18 | 8 | 45 | 11 | 13 | 21 |
| 8 | 49 | 68 | 72 | 14 | 12 | 43 | 10 | 12 | 20 |
| 9 | 53 | 93 | 57 | 8 | 4 | 42 | 9 | 17 | 17 |
| 10 | 53 | 90 | 59 | 12 | 2 | 46 | 11 | 12 | 18 |
| 11 | 59 | 106 | 56 | 17 | 15 | 39 | 19 | 24 | 19 |
| 12 | 75 | 136 | 55 | 7 | 11 | 45 | 19 | 12 | 19 |

*Data on participant role(s) were not collected for Summits 1–3.

[a]Summit 1 (2008) to Summit 12 (2019)

[b]Percentages may total more than 100% because participants could select more than one role

[c]Examples of "Other" participant roles include: representatives of non-profit organizations, health care professionals who serve individuals with dementia but who did not endorse "working directly with people with dementia in rural areas," provincial Ministry of Health directors, representatives of grant funding bodies, Alzheimer Society staff and leadership (all of whom attend yearly).

The open-ended evaluation questions were designed to explore various facets of the Summit each year, including engagement in the research, organizational aspects, specific Summit elements, ideas for future Summits, and impacts. The linkages between the themes identified in the stakeholder and researcher data, and the principles and impacts of stakeholder engagement defined in the literature, are reported in **Table 3**.

The thematic analysis of stakeholder data identified five themes across the open-ended questions: hearing diverse perspectives, building connections and relationships, collaborating for change, developing research and practice capacity, and leaving recharged. Examples of illustrative quotations are reported for each theme. As seen in **Table 3**, these themes can be linked to the principles of stakeholder engagement.

**Hearing diverse perspectives.** Engaging a broad range of stakeholders is a central tenet of stakeholder engagement. A key theme was appreciation for the variety of perspectives represented by Summit participants and speakers, which provided a rich environment that was different from typical meetings and conferences. Stakeholders found it stimulating to meet and learn from people with diverse backgrounds, all with a stake in improving rural dementia care.

*"Being in a room with family members, researchers, decision-makers, and practitioners engaged in knowledge translation and exchange."* [Summit 3]

*"Interprofessional collaboration. Usually I work in my own silo as a discipline. This opened my eyes to all the people involved and working towards these initiatives, that I never knew of before."* (Summit 11)

 Stakeholder and researcher perspectives of an engagement strategy for a rural dementia research program

**Table 3. Alignment of stakeholder and researcher themes with published stakeholder engagement principles and impacts.**

| Stakeholder Engagement Principles* | Stakeholder Themes | | | | | Researcher Themes | | | | |
|---|---|---|---|---|---|---|---|---|---|---|
| | Hearing diverse perspectives | Building connections and relationships | Collaborating for change | Developing research & practice capacity | Leaving recharged | Impact on development as researcher | Understand stakeholder needs | Inform research design | Deepen commitment to rural dementia research | Build culture of engagement |
| Relationship based on trust, respect | | | √ | | | | | | | √ |
| Ensure diversity, inclusiveness | √ | | | | | | | | | √ |
| Share decision-making | | | √ | | | | | | | √ |
| Engagement across research phases | | | √ | | | | | √ | | |
| Benefits for researchers and stakeholders | √ | √ | √ | √ | √ | √ | √ | √ | √ | √ |
| Builds stakeholder capacity for meaningful engagement | | | | √ | | | | | | |
| Bi-directional exchange of skills, knowledge | | | | √ | | | √ | √ | | |
| **IMPACTS of SE** | | | | | | | | | | |
| **For Researchers** | | | | | | | | | | |
| Increased capacity/skills re: collaborative research | | | | √ | | √ | | | | |
| Enhanced motivation for the project | | | | | √ | | | | √ | |
| **For Stakeholders** | | | | | | | | | | |
| Increased capacity r/t research processes | | | | √ | | √ | | | | |
| More positive attitude and understanding of research | | | | √ | | | | | √ | |
| Better access to information re: managing specific diseases | | | √ | √ | | | | | | |
| Personal benefits (confidence, empowered, feel valued, sense of accomplishment, extended social network) | | | √ | √ | √ | | √ | | | |

*(Continued)*

**Table 3.** (Continued)

| Stakeholder Engagement Principles* | Stakeholder Themes | | | | | | Researcher Themes | | | | |
|---|---|---|---|---|---|---|---|---|---|---|---|
| | Hearing diverse perspectives | Building connections and relationships | Collaborating for change | Developing research & practice capacity | Leaving recharged | | Impact on development as researcher | Understand stakeholder needs | Inform research design | Deepen commitment to rural dementia research | Build culture of engagement |
| **For Relationship** | | | | | | | | | | | |
| Mutual respect & understanding of language, needs, constraints | | | | √ | | | | √ | | | √ |

*from reviews by Hoekstra et al. [8] and Harrison et al. [16] and Patient Centered Outcomes Research Institute (PCORI) Engagement Principles (Sheridan et al. [19]).

**from review by Hoekstra et al. [8]

Many participants also commented on the variety of topics presented by researchers, front-line providers, other stakeholders, and those living with dementia.

*"Good balance of research-focused, clinical, and front-line stakeholders' presentations—and good representation from all groups."* (Summit 4)

*"The diversity of presentation is really important to get all perspectives on the issues of dementia care."* (Summit 4)

Hearing the personal "real-life" stories of persons living with dementia and their families helped participants better understand what it is like to live with dementia and what is needed to support living well with dementia in a rural context. These stories were very engaging, and inspired attendees to make changes to improve care and supports.

*"Story telling comments—brings humanity to research."* (Summit 5)

*"I appreciated that the emphasis was beyond research findings, to include meaningful group discussion and lived experiences."* (Summit 11)

**Building connections and relationships.** The Summit agenda, including the timing of sessions and networking breaks, encourages interaction between and among stakeholders and researchers. Participants stated that the opportunity for networking with others with similar passions about rural dementia care was an important feature of Summit that was different from other meetings.

*"Highly relevant networking. Dementia specific focus is great and unlike other conferences."* (Summit 10)

*"An opportunity to network with an interdisciplinary and multi-stakeholder group of people who are focused on dementia care within a rural context."* (Summit 11)

Participants appreciated the opportunity to meet new people and re-connect with those they had met at previous Summits. Making a personal connection made it possible for them to follow up after Summit.

*"Opportunity to meet new people and re-connect with others who are passionate about dementia care."* (Summit 4)

*"The networking over time has allowed me to consult with confidence when there are questions in dementia care."* (Summit 10)

Other benefits of networking included sharing information, learning about resources and programs available in other places, and improving the care provided to their clients.

*"It's a special opportunity to learn about local initiatives and connect with people in the province."* (Summit 10)

*"The opportunity to appreciate the wealth of knowledge in our province—great for relationship and partnership opportunities."* (Summit 11)

**Collaborating for change.**   A highlight for many participants was the interactive small-group session focused on engaging stakeholders in shaping a RaDAR project. Participants valued the opportunity to play a role in the research process and help address challenges in rural dementia care.

"*A chance to feel like I have a voice to getting some things dealt with/looked at.*" (Summit 9)

*"Opportunities to brainstorm priorities for research—great discussion."* (Summit 11)

Assignment to the small groups was made with the goal of maximizing stakeholder diversity at each table. Stakeholders had more opportunities to speak and listen to other stakeholders in these sessions, which provided new and deeper insights into both the impact of dementia and potential strategies for addressing challenges.

*"The group discussion provided an excellent opportunity to speak to and learn from others in different health care positions and regions."* (Summit 8)

*"Just the break-out [small group] sessions alone opens up your perspective to so much more than is likely when working in silos as we often do. . . taking the time to connect with the broader community is essential."* (Summit 11)

The Summit as a whole, and the small group sessions in particular, provided opportunities for engagement and being part of an ongoing, collective effort to bring about positive change. There was a sense of reciprocity, both contributing and receiving something in return. This feeling of community and working toward shared goals over time was mentioned by many stakeholders as the aspect of Summit they valued most.

*"So much value in this [small group work] session. Excellent work and a great comfort to know that we are addressing gaps in our healthcare system."* (Summit 5)

*"[The Summit provides] a sense of community and collaboration–commonality."* (Summit 12)

Several participants commented on how the Summit sustains engagement by providing a safe, open, and inclusive environment where all voices are valued and respected.

*"I appreciated how you bring together so many people with different life and education experiences and you are willing to listen and incorporate all voices."* (Summit 9)

*"The variety of presentation topics, the mix of participants, the openness to feedback and recommendations."* (Summit 9)

**Developing research and practice capacity.** The Summit shared information about research being planned and conducted by the RaDAR team, other researchers in the province and Canada, and internationally, which was not easily available to most participants. The exposure to research and the ability to interact face-to-face with researchers increased stakeholders' understanding of the research process and their comfort with actively participating in it.

*"This was a great way to see how the research comes about and how studies are done."* (Summit 2)

*"I appreciated being able to see all the work/research being done and to ask questions and get clarification on the work being done."* (Summit 11)

Participants valued learning about the links between research and practice, and hearing examples of how research has been translated, particularly in rural settings. This provided inspiration and practical strategies for adapting innovations to their own communities.

*"[I gained] more insight on other research happening in the province, and in particular how this has translated into actions (create support services)."* (Summit 9)

*"It provided me with a stellar example of knowledge translation and I had several excellent conversations about rural needs."* (Summit 9)

Participants appreciated the unique rural-specific focus of the Summit. They gained a better understanding of the barriers in rural dementia care from a variety of perspectives, but also the possibilities and opportunities to explore new ways of solving challenges in their own practice through research and application of existing knowledge.

*"The emphasis on dementia and rural and remote care. These are not always a priority in other events."* (Summit 10)

*"The focus of persons living with dementia rurally is essential to striding forward with dementia. I love hearing about ideas, interventions, models of delivery, successes."* (Summit 11)

In addition to building capacity to engage in research, the Summit also helped build stakeholder capacity to effect change in their communities. Many participants reported that unlike a typical conference, at Summit they learned about specific tools, resources, programs, and models of care that they could readily apply in their own rural practice or take back to their staff to help build capacity in front-line care providers. Some participants reported that their involvement in Summit had influenced policy decisions.

*"I believe the knowledge gained by me and brought back to the workplace is unmeasurable."* (Summit 10)

*"Concrete examples of immediate actions/info we can take back and use."* (Summit 10)

Participants found it encouraging to hear about locally-developed interventions and programs that addressed real problems in rural communities and that they could use to improve the care they provided.

*"New information to use everyday and to improve on my knowledge and to better serve clients."* (Summit 10)

*"It is local. It is encouraging that things can be done. It is practical in having fixes of real issues."* (Summit 10)

**Leaving recharged.**   The last theme was related to the positive influence of the Summit on attendees' attitudes toward dementia care and their work. Participants used terms such as being rejuvenated, revitalized, energized, and re-charged by the opportunity to engage with other stakeholders and researchers committed to improving dementia care.

*"We all leave with new knowledge and re-motivated to further our work with dementias."* (Summit 4)

*"I gain a refreshed passion for what I do each year—rejuvenates me."* (Summit 10)

The Summit provided new ideas to take away and stimulated participants to think about applications in their own work, which they found exciting and motivating.

*"I find the Summit very empowering and a great re-charge to go back to your work."* (Summit 9)

*"This Summit leaves me energized to go forward in dementia care."* (Summit 12)

Finally, connecting with others and learning about new initiatives in dementia care gave attendees hope that care and quality of life could be improved for individuals living with dementia and their formal and informal caregivers.

*"It was very encouraging to hear from caregivers that the poster session gave them hope and understanding that there is lots going on to improve dementia care."* (Summit 11)

*"A genuine hope that people facing dementia will have a better experience/a safer experience."* (Summit 11)

## RaDAR research team perspectives

Five themes were identified in the research team data. The linkages of these themes with stakeholder engagement principles and impacts are shown in **Table 3**.

**Impact on their development as a researcher.**   Trainees and early career researchers commented that being involved in the Summit has influenced them as researchers, including changing their perspective on how to work with stakeholders and the value of CBPR.

*"Has changed how I think about working with stakeholders—really appreciate the value of it more."*

*"The Summit has helped shape me as a researcher, and has helped to immerse me in real-life issues/practice."*

The Summit provided an opportunity to gain new knowledge and skills related to CBPR and direct experience with a successful model of stakeholder engagement.

*"Being involved in the Summit has provided me a great model of how to bridge gaps between people working in different sectors of dementia care research."*

*"As a trainee, this has allowed me to learn more about how to do research that is community-based."*

Early career investigators also benefited from the established structure of the Summit by being mentored by senior researchers and receiving feedback on their research from people with dementia, families, health care providers, and other stakeholders.

*"The established network of the Summit group would take a decade to develop, and it is an asset that is provided to us. . . this model is already well-honed and operates for us. This is invaluable to an early-career researcher."*

**Understanding stakeholder needs and priorities.**   A key theme was the advantage of understanding what was important to stakeholders and why, so researchers could ensure that their research was relevant and useful.

*"I find it very beneficial and thought-provoking to hear stakeholders' perspectives on what I have produced as related (or sometimes not) to their priorities and lived experiences. . . is what I am presenting important/of value to them? How so? Why? What can I do differently to align with their priorities?"*

*"I believe their input has made some of the grants more successful because the engagement is meaningful and the research is modified to meet the stakeholders' feedback."*

Engaging with stakeholders grounded researchers in the realities of rural dementia care and its impact on those it was designed to help.

*"Summit keeps us rural-centric, and keeps us out of our "ivory tower."*

*"I see my research in the context that it may be used. . . not just creating knowledge in a vacuum."*

**Informing research design and methods.**   The learning that researchers gain at Summit directly influences their research. The small group discussions have been critical in identifying research priorities, developing new projects, and understanding adaptations to ensure feasibility in rural settings (**S2 Table**). Assessing compatibility with existing resources, identifying methodological challenges and solutions, and guiding ongoing research were also positive engagement outcomes.

*"[Summit] really helped me understand location adaptations to our rural settings and how resources in the field would impact the methods of planned studies. The rural settings are all so different."*

*"I really got a sense of barriers to adaptations that I hadn't expected. . . all the mitigation strategies in the grant application came from discussions at the Summit."*

Engaging with stakeholders provides momentum and direction to the research, by showing the way forward for a project and providing encouragement to try new approaches.

*"Collaborating with patients/families, providers, decision-makers pushes the research—sometimes in different directions . . . sometimes faster. . . Knowing that an idea is supported or valued by patients and families can provide courage to take a research risk or move in a direction we might not have thought of. It can also help us let go of ideas."*

The ongoing nature of the Summit allows researchers to engage with stakeholders across the trajectory of a project and provides an incentive to be ready for Summit each year in order to capitalize on the opportunity to interact with stakeholders.

*"We work toward the Summit as a goal of having research to present and share for feedback (completed, in progress, ideas in development)."*

**Increasing commitment to rural dementia care research.** RaDAR trainees and new investigators reported that connecting with stakeholders who shared a common interest and learning about the value of their research to stakeholders provided validation and motivation, reinforcing their commitment to rural dementia research.

*"My Summit experience really helped me to take the leap into a PhD–I felt like I could do meaningful work that would be of value. And each year I would feel encouraged to work hard."*

*"Research can be a lonely pursuit in some ways, and the impact can be far-removed from the researcher experience. . . The encouragement and support given by [stakeholders] helps to maintain the long-term focus of the research."*

Hearing from stakeholders about their experiences and concerns helped trainees to see the relevance and application of their research, deepening their connection to the research.

*"I learn other perspectives, but I also build empathy for the different stakeholder groups. I find myself caring a lot more about rural dementia research on a personal level."*

*"[Summit] has allowed me to see the real potential impact of my research. . . which made my research seem more relevant and applied. It gave my research more meaning."*

**Building a culture of engagement.** The last theme relates to RaDAR team members' observations that a culture of engagement has evolved over time because of ongoing contacts and an emphasis by RaDAR leadership and members on relationship development.

*"There is a supportive aspect to the Summit that is continual. Relationships are fostered and communication (formally and informally) happens regularly. This is all underscored by leadership–without that, Summit and the collaborations wouldn't happen. Someone has to have a vision and guide the process."*

*"It's building a culture and a community—a community-based culture has been built through these relationships over time."*

Ongoing interaction with the RaDAR team over the years has built the capacity of stakeholders to engage with research and to contribute their experience and knowledge in an atmosphere of trust and openness.

*"Summit is about building capacity and empowerment of stakeholders. . . word is out it's a safe place to bring up ideas and share your voice."*

*"The long-term nature helps to develop true relationships. . . Over time, this helps in terms of trust. Summit becomes a safe place to test out new ideas and brainstorm or problem solve in a trusting, safe environment."*

Stakeholders who return to Summit see that their ideas are used, which also feeds back into greater engagement and trust.

*"There is a lot of lip service in engagement research, but Summit seems to create a sense of being part of a "thing" that is dynamic and moves forward. . . one year participants provide ideas. . . and the next year we're reporting back on how those comments have already been used to shape an application for funding."*

## Challenges associated with implementing and sustaining the Summit

As stakeholder and researcher findings demonstrate, the Summit has been an effective engagement strategy, but implementing and sustaining the Summit has not been without challenges. Planning each year often requires balancing conflicting stakeholder recommendations. For example, participants have suggested representation from additional sectors, but also recommend keeping the Summit small and intimate in contrast with regular conferences. Increasing the size of Summit also conflicts with the need to manage costs of hosting the in-person Summit ($15,000 CDN in 2019). We need to balance requests for additional session topics with keeping the length of the Summit reasonable, as time away from work is difficult for some stakeholders, who may also have long travel times. Some stakeholders have noted that the day can feel rushed, while others appreciate the range of content and keeping sessions on time. Planning the session start and end times requires balancing requests for earlier vs. later times depending on stakeholders' location and circumstances. Choosing an ideal time of year is complicated by seasonal weather issues that make driving dangerous and by the wish to avoid conflicts with other academic conferences. It is an ongoing challenge to attract physicians and persons living with dementia who are willing and able to travel to Summit. Although our communications about the Summit describe the engagement focus, we occasionally receive comments that suggest people are expecting a more traditional conference and thus Summit has not met their expectations.

## Discussion

This paper addresses gaps in the stakeholder engagement literature by describing a strategy for an ongoing research program focused on rural dementia care. Themes identified in the stakeholder and researcher data reflect the key principles of stakeholder engagement: development of relationships based on trust, ensuring diversity of stakeholder representation, shared

decision-making, engagement across the research process, benefits for both researchers and stakeholders, and two-way exchange of knowledge [8, 16]. However, findings from the current study also expand our understanding of how these engagement principles operate in an ongoing research program, and their short and long-term impacts.

Our findings support stakeholder outcomes hypothesized and reported in the literature, including increased learning of research skills and knowledge of the topic area, increased capacity to help the communities they serve, building professional relationships, increased confidence, and sense of personal achievement, and feeling valued and involved [5, 36, 37]. Findings not reported elsewhere include the importance of the rural focus, the value stakeholders place on connecting with and learning from other stakeholders from diverse backgrounds, and how participation in the Summit revived their commitment to and engagement in dementia work. The evaluations also highlighted how ongoing exposure to a wide range of research methods and findings helped build stakeholder comfort and capacity to engage in the research process; similarly, learning about current best practices and programs in rural dementia care developed stakeholder practice capacity to support change in their communities. Understanding stakeholders' motivations for participating in a community-academic partnership and their definitions of success have been found to be key to sustaining the partnership [38]. The current analysis identified factors that stakeholders found meaningful about being involved in the Summit and that motivated their ongoing participation.

RaDAR researchers' perspectives of the Summit as an engagement strategy are consistent with other studies examining benefits of engagement, including receiving stakeholder direction into research topics, questions, and methods [1, 2, 4], which has improved the quality and relevance of RaDAR research. Our findings align with those reported in the literature, that many of the impacts on research result from what researchers learn from stakeholder engagement, including new research ideas, understanding issues that are significant to stakeholders, identifying potential problems and how to avoid them, and increased confidence and motivation to pursue a project [39]. Other outcomes of engagement, particularly for RaDAR trainees and early career researchers, were development of skills for collaborating with stakeholders and appreciation of the value of CBPR approaches. Our results are reflected in a review of participatory research approaches [9], which found that over time both stakeholders and researchers gained capacity and competence, resulting in benefits for the specific program, for other community activities, and for personal and professional development of both groups.

The fact that the Summit has been held annually for over a decade and many stakeholders attend regularly has helped to foster trust between researchers and stakeholders, and among stakeholders. Sustained interaction is needed to build trustful relationships [40, 41], which is the basis of success for CBPR partnerships [37]. Slunge et al. [20; pg. 22] note that one objective of developing an engagement strategy for a research group or ongoing program (vs. a specific project) is "to create a culture of stakeholder interaction" and a supportive infrastructure for engagement. The RaDAR team has actively endeavored to create a respectful, safe environment for engagement. A culture of inclusivity, respect, and trust has developed over time as Summit stakeholders recognize RaDAR's commitment to hearing all voices and to improving rural dementia care, and our willingness to continue investing research funding and time in organizing the event.

By continuously addressing stakeholder needs and interests, the Summit has evolved to serve a broader function as a knowledge mobilization hub to provide access to information that is otherwise not easily available. In rural areas of the province there are few opportunities to connect with and learn from others with similar interests, and to learn about new programs, services, and research directly relevant to stakeholders' practice. We have responded by ensuring that the Summit planning focuses on the elements identified as important to stakeholders,

which contributes to relationship development and trust, and the effectiveness and sustainability of the Summit.

Findings from the current study suggest that the Summit has developed into a community of practice (CoP) in rural and remote dementia care. A CoP refers to a group of people who share a common concern, who come together regularly to create and share knowledge [42]. Pyrko et al. [43] state that mutual engagement is an essential element of a CoP, and that the process of "thinking together" by people with different personal knowledge of the topic is what brings CoPs to life. A CoP cannot be "set up" but develops organically as a place for having regular conversations that provide immediate value, such as providing solutions to members' problems or practical tools relevant to their work [42, 43]. The Summit brings together stakeholders from different backgrounds who share a common interest, to learn about innovations in rural dementia care, and to work together on research projects addressing rural dementia care issues. Although it was not our original intent to create a CoP, activities included in the Summit to encourage interaction and engagement, and to address stakeholders' desire to hear about other rural dementia programs and services, in addition to RaDAR projects, have supported the emergence of a CoP.

In a review of the benefits of engagement in participatory research, Jagosh et al. [9] found that the partnerships created systemic changes and activities beyond project goals because stakeholders were integrated into local contexts and could effect change. This description fits with many Summit stakeholders as well. In rural settings front-line providers, directors, and managers often hold several roles and have influence in multiple spheres. Summit participants described using or sharing the ideas learned at Summit to improve care. Other studies have observed similar impacts of engagement, including Silvestre et al. [44] who found that a 20-year community advisory board for a research program on AIDS resulted in scientific contributions, but more importantly led to significant social change by improving public health services. Cook et al. [1] found that building ongoing networks during the research led to enhanced capacity for change. It has been noted that longitudinal studies of engagement models are needed to see their potential for transforming research programs, influencing researchers' career trajectories, and changing the culture of research [7]. As the Summits continue we will be able to further track these researcher outcomes as well as the impact on stakeholders and improved service delivery and system changes. In the future we may also be able to compare the virtual vs. in-person Summit experience, as we hold virtual Summits due to the pandemic. Based on two virtual Summits to date, we found that participants appreciated the greater accessibility due to not having to travel or take time from work and home, as well as the flexibility to view pre-recorded posters and sessions at times convenient to them. However, many commented that they missed connecting and networking face-to-face, building relationships, and problem-solving together.

A study strength is the availability of data over 12 years with the majority of questions continued each year, and the inclusion of stakeholder and researcher perspectives. A potential limitation is the broad nature of the evaluation data collected each year, which is intended to provide direction for future Summit planning and was not exclusively focused on exploring or evaluating the Summit as an engagement strategy. However, the data include many references relevant to engagement, highlighting its salience to stakeholders. Questions about impact were added for Summits 9–12 which provide rich data on stakeholders' perspectives of engagement. The lack of some demographic data for stakeholders (e.g., sex and gender) is another limitation, as we are unable to determine if sex or gender influence perspectives or experiences. Based on the development of trust over the years we recently added sex and gender questions. We also plan to identify and record individuals living with dementia and caregivers separately. We previously had one category for both because of initial concerns that identifying as

someone with dementia may be stigmatizing, but we recognize the importance of legitimizing and validating their separate perspectives and input. Finally, there is a potential for bias in the analyses, which were conducted by the authors, most of whom are RaDAR researchers engaged in the Summit. This was mitigated by involving multiple team members and two non-RaDAR members in independently analyzing the data. Some participants, particularly those new to Summit, may have been uncomfortable making negative comments. Despite this concern, we did receive constructive feedback such as recommendations to ensure sessions stay on time, to allow more time for discussion and feedback, to expand to additional stakeholder groups, and to include even more input from those with lived experience of dementia.

## Conclusion

In this paper we described the development and implementation of an annual engagement event for an ongoing rural dementia research program, and stakeholder and researcher perspectives. Our findings show that the Summit is an effective engagement strategy for the RaDAR research program, with mutual benefits for stakeholders and researchers that go beyond research impacts. The Summit works as an engagement strategy and has been sustained because both groups find value in their participation. Stakeholders gain new knowledge from other stakeholders and researchers, make valuable connections, contribute to improving rural dementia care, develop research and practice capacity, and leave inspired and rejuvenated in the work. Researchers also benefit, gaining skills in stakeholder engagement, understanding needs and priorities for research and how to design their research to address these, and reinforcing their commitment to this research area. By actively responding to stakeholder needs versus a limited focus on our research needs, we have nurtured a reciprocal relationship that has supported long-term engagement and commitment, resulting in growing momentum each year as partnerships strengthen and new stakeholders become involved.

## Supporting information

**S1 Table. Summit components and their purpose.**
(PDF)

**S2 Table. Example of small group engagement sessions.**
(PDF)

**S3 Table. How the Summit differs from a typical conference.**
(PDF)

**S1 File. Summit interactive small group sessions.**
(PDF)

**S2 File. Stakeholder data.**
(PDF)

**S3 File. Research team data.**
(PDF)

## Acknowledgments

We would like to thank Summit participants for completing the evaluations each year, providing valuable direction to the RaDAR research program and guiding the planning of future Rural Dementia Summits.

## Author Contributions

**Conceptualization:** Debra Morgan, Julie Kosteniuk, Megan E. O'Connell, Norma J. Stewart, Andrew Kirk, Allison Cammer, Vanina Dal Bello-Haas, Duane P. Minish, Valerie Elliot, Melanie Bayly, Amanda Froehlich Chow, Joanne Bracken, Edna Parrott, Tanis Bronner.

**Data curation:** Debra Morgan, Julie Kosteniuk, Duane P. Minish, Valerie Elliot.

**Formal analysis:** Debra Morgan, Julie Kosteniuk, Valerie Elliot.

**Funding acquisition:** Debra Morgan, Julie Kosteniuk.

**Investigation:** Debra Morgan, Julie Kosteniuk, Duane P. Minish, Valerie Elliot.

**Methodology:** Debra Morgan, Julie Kosteniuk.

**Project administration:** Debra Morgan.

**Resources:** Debra Morgan, Julie Kosteniuk, Megan E. O'Connell.

**Software:** Debra Morgan, Valerie Elliot.

**Supervision:** Debra Morgan.

**Validation:** Debra Morgan, Julie Kosteniuk, Duane P. Minish, Valerie Elliot, Melanie Bayly.

**Visualization:** Julie Kosteniuk, Duane P. Minish, Melanie Bayly.

**Writing – original draft:** Debra Morgan.

**Writing – review & editing:** Debra Morgan, Julie Kosteniuk, Megan E. O'Connell, Norma J. Stewart, Andrew Kirk, Allison Cammer, Vanina Dal Bello-Haas, Duane P. Minish, Valerie Elliot, Melanie Bayly, Amanda Froehlich Chow, Joanne Bracken, Edna Parrott, Tanis Bronner.

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
