## [Decision Letter · Decision Letter 0]

3 May 2022

PONE-D-21-34146A stakeholder engagement strategy for an ongoing research program in rural dementia care: Stakeholder and researcher perspectivesPLOS ONE

Dear Dr. Morgan,

Thank you for submitting your manuscript to PLOS ONE. After careful consideration, we feel that it has merit but does not fully meet PLOS ONE’s publication criteria as it currently stands. Therefore, we invite you to submit a revised version of the manuscript that addresses the points raised during the review process.

Bothe the reviewers acknowledged the overall quality of the paper and the relevance of the topic. However, they pointed to some important revisions, including the reduction of details that make the paper difficult to read. The paper need stylistic revision, including a consideration for the recommendations at the more scientific level. 

We look forward to receiving your revised manuscript.

Kind regards,

Sara Rubinelli

Academic Editor

PLOS ONE

Journal Requirements:

4. Thank you for stating the following in the Acknowledgments Section of your manuscript: "Funding for this research was provided by the Bilokreli Family Trust Fund, and the Saskatchewan Health Research Foundation through a partnership with the Canadian Institutes of Health Research, in support of the Canadian Consortium in Neurodegeneration in Aging (CCNA, https://ccna-ccnv.ca/) (grant number 3431)."

"The authors received no specific funding for this research.  The reported stakeholder engagement strategy (Rural Dementia Summit) is supported by the Bilokreli Family Trust Fund awarded to the first author DM. The funders had no role in study design, data collection and analysis, decision to publish, or preparation of the manuscript."

7. Please amend your list of authors on the manuscript to ensure that each author is linked to an affiliation. Authors’ affiliations should reflect the institution where the work was done (if authors moved subsequently, you can also list the new affiliation stating “current affiliation:….” as necessary).

Reviewers' comments:

Reviewer's Responses to Questions

**Comments to the Author**

1. Is the manuscript technically sound, and do the data support the conclusions?

Reviewer #1: Partly

Reviewer #2: Yes

2. Has the statistical analysis been performed appropriately and rigorously? 

Reviewer #1: N/A

Reviewer #2: N/A

3. Have the authors made all data underlying the findings in their manuscript fully available?

Reviewer #1: Yes

Reviewer #2: Yes

4. Is the manuscript presented in an intelligible fashion and written in standard English?

Reviewer #1: Yes

Reviewer #2: Yes

5. Review Comments to the Author

Reviewer #1: This is generally a well-written manuscript that reports on the laudable goal of stimulating participatory research involving a broad range of stakeholders involved in rural dementia research and care over many years.

My major concern is that a substantial proportion of the manuscript reads like an organisational report, rather than the product of systematic research. The paper would benefit from a clear early section about the research methodology and how this was designed to answer specific research questions, with some justification/rationale around the mixed methods approach.

I would suggest the authors have a close look at sections of the paper such as Table 1 and 2 as to whether they fit in a research report. Table 3 may benefit with linkages to relevant research papers as well as outcomes that affect delivery of care in a rural setting.

I am also concerned that some of the data presented, particularly evaluation data, was deemed exempt from ethics committee consideration. Was this because this was for internal evaluation purposes initially and/or were participants aware that data may be reported?

It would be useful for the authors to consider issues of bias and motivation in their reporting. Clearly, those attending would have been highly motivated in this area? In addition, it should be made clear whether it is RaDAR team that comprises the authorship of the article. Were any of the analyses conducted by researchers who were not engaged in RaDAR or the summit? I note the absence of any negative themes or quotes, which may well be the case, but it would be good to hear more about this paper’s research approach to reduce bias.

Reviewer #2: The work reported in this paper is very informative and of high relevance. The paper is very well and clearly written. However, I would strongly recommend the authors to review the paper and reduce it substantially in length. Please consider a more concise writing style and review whether the amount of detail provided in the current version of the manuscript is really necessary for the optimal communication of the message. As I mention, given the importance of the message, it would be best if the length and amount of detail provided does not discourage potentially interested readers.

6. PLOS authors have the option to publish the peer review history of their article (what does this mean?). If published, this will include your full peer review and any attached files.

Reviewer #1: No

Reviewer #2: No

---

## [Author Response · Author response to Decision Letter 0]

17 Aug 2022

RESPONSE TO REVIEWS (this information also provided as a separate file that I uploaded with the revised manuscript).

The authors would like to thank the reviewers for their thoughtful and helpful comments on the manuscript. We appreciate the opportunity to revise the paper and hope that we have addressed their recommendations to their satisfaction. 

A. Journal Requirements. When submitting your revision, we need you to address these additional requirements:

This has been done.

2. Please provide additional details regarding participant consent. …If the need for consent was waived by the ethics committee, please include this information.

This has been done. Consent was waived by the ethics committee because the study was deemed program evaluation. However, Summit participants were informed that the evaluations could be used for publications. The evaluations were completed anonymously.

3. We note that the grant information you provided in the ‘Funding Information’ and ‘Financial Disclosure’ do not match. When you resubmit, please ensure that you provide the correct grant numbers for the awards you received in the ‘Funding Information’ section.

This was done. The information is included in the cover letter as requested.

4. We note that you have provided funding information that is not currently declared in your Funding Statement. Funding information should not appear in the Acknowledgments or other areas of your manuscript. Please remove any funding-related text from the manuscript and let us know how you would like to update your Funding Statement. Please include your amended statements within your cover letter.

We have removed funding information from the Acknowledgements and included it in the cover letter.

5. In your Data Availability statement, you have not specified where the minimal data set underlying the results can be found. Upon re-submitting, please upload your study’s minimal underlying data set as either Supporting Information files or to a stable, public repository and include the relevant URLs, DOIs, or accession numbers within your revised cover letter. We will update your Data Availability statement.

The original submission stated that the data were provided as Supporting Information Files attached to the submission. This information is included in the cover letter. 

7. Please amend your list of authors on the manuscript to ensure that each author is linked to an affiliation. 

All authors have an affiliation except for a family caregiver (EP) who does not have an affiliation as she is retired and not linked to any organization.

B. Reviewers' comments:

Reviewer #1: 

This is generally a well-written manuscript that reports on the laudable goal of stimulating participatory research involving a broad range of stakeholders involved in rural dementia research and care over many years. 

My major concern is that a substantial proportion of the manuscript reads like an organisational report, rather than the product of systematic research. The paper would benefit from a clear early section about the research methodology and how this was designed to answer specific research questions, with some justification/rationale around the mixed methods approach.

Thank you for these comments. We agree that a statement about the methodology should have been included and have added this information at the beginning of the Methods section (Page 8, starting line 195). The more organizational aspects of the Summit in tables have been moved to Supporting Information files (see below).

I would suggest the authors have a close look at sections of the paper such as Table 1 and 2 as to whether they fit in a research report. Table 3 may benefit with linkages to relevant research papers as well as outcomes that affect delivery of care in a rural setting.

Based on this suggestion, and the recommendation of Reviewer 2 to remove some of the detail and reduce the length of the manuscript, we have removed Tables 1 – 3 and Box 2 from the manuscript. These tables are now included as Supporting Information files S1 (Summit components and their purpose), S2 (Example of Summit small group engagement sessions), S3 (how Summit differs from a typical conference), S4 (Interactive small group sessions focus and outcomes). Because of the gap in the engagement literature regarding the structure and processes of engagement strategies for ongoing research programs, we feel it is important that readers have access to this information. These structure and processes evolved in response to stakeholder feedback and are linked to stakeholder satisfaction with the Summit and the outcomes reported in the thematic analysis. In Table S2 the term “outcome” in one of the headings was changed to “purpose” to more accurately reflect the content of that column.

I am also concerned that some of the data presented, particularly evaluation data, was deemed exempt from ethics committee consideration. Was this because this was for internal evaluation purposes initially and/or were participants aware that data may be reported?

Yes the University ethics committee deemed the evaluations exempt from ethics review because it was collected for program improvement. Participants were informed that the evaluations may be used for reports and publications.

It would be useful for the authors to consider issues of bias and motivation in their reporting. Clearly, those attending would have been highly motivated in this area? In addition, it should be made clear whether it is the RaDAR team that comprises the authorship of the article. Were any of the analyses conducted by researchers who were not engaged in RaDAR or the summit? I note the absence of any negative themes or quotes, which may well be the case, but it would be good to hear more about this paper’s research approach to reduce bias.

Summit participants were invited based on their interest and expertise in dementia care, or were self-identified individuals who requested to attend, so they were highly motivated. The authors are RaDAR researchers, except for a family caregiver and rural nurse practitioner who have both had a long association with the RaDAR team and are regular Summit attendees. Many Summit participants attend every year and may be more comfortable reporting where improvements could be made, but it is possible that new participants would be less likely to make negative comments. Most of the “negative“ findings were related to organizational aspects such as Summit dates or start and end times, keeping to schedule so there is time for discussion, suggesting additional stakeholder groups to invite, etc, and these are noted in the Challenges section. Potential bias in the analysis was mitigated by the team approach to the analysis of themes, and inclusion of non-RaDAR members. 

Reviewer #2: 

The work reported in this paper is very informative and of high relevance. The paper is very well and clearly written. However, I would strongly recommend the authors to review the paper and reduce it substantially in length. Please consider a more concise writing style and review whether the amount of detail provided in the current version of the manuscript is really necessary for the optimal communication of the message. As I mention, given the importance of the message, it would be best if the length and amount of detail provided does not discourage potentially interested readers.

Thank you for your comments, and the recommendation for a more concise paper. As noted under Reviewer 1’s second point above, we have removed 8 pages of detail by relocating Tables 1-3 and Box 1 from the text to Supporting Information Files and removing some text throughout the manuscript. We agree that these changes make the paper more accessible, while still providing interested readers with the specifics regarding Summit organization and content in the optional Supporting Information files.

---

## [Decision Letter · Decision Letter 1]

4 Sep 2022

A stakeholder engagement strategy for an ongoing research program in rural dementia care: Stakeholder and researcher perspectives

PONE-D-21-34146R1

Dear Dr. Morgan,

We’re pleased to inform you that your manuscript has been judged scientifically suitable for publication and will be formally accepted for publication once it meets all outstanding technical requirements.

Kind regards,

Sara Rubinelli

Academic Editor

PLOS ONE

Additional Editor Comments (optional):

Reviewers' comments:

Reviewer's Responses to Questions

**Comments to the Author**

1. If the authors have adequately addressed your comments raised in a previous round of review and you feel that this manuscript is now acceptable for publication, you may indicate that here to bypass the “Comments to the Author” section, enter your conflict of interest statement in the “Confidential to Editor” section, and submit your "Accept" recommendation.

Reviewer #1: All comments have been addressed

2. Is the manuscript technically sound, and do the data support the conclusions?

Reviewer #1: Yes

3. Has the statistical analysis been performed appropriately and rigorously? 

Reviewer #1: Yes

4. Have the authors made all data underlying the findings in their manuscript fully available?

Reviewer #1: Yes

5. Is the manuscript presented in an intelligible fashion and written in standard English?

Reviewer #1: Yes

6. Review Comments to the Author

Reviewer #1: (No Response)

7. PLOS authors have the option to publish the peer review history of their article (what does this mean?). If published, this will include your full peer review and any attached files.

Reviewer #1: **Yes: **James Vickers

---

## [Editor Report · Acceptance letter]

12 Sep 2022

PONE-D-21-34146R1 

A stakeholder engagement strategy for an ongoing research program in rural dementia care:  Stakeholder and researcher perspectives 

Dear Dr. Morgan:

I'm pleased to inform you that your manuscript has been deemed suitable for publication in PLOS ONE. Congratulations! Your manuscript is now with our production department. 

Kind regards, 

on behalf of

Dr. Sara Rubinelli 

Academic Editor

PLOS ONE